# StepProof: Step-by-step verification of natural language mathematical proofs

## Abstract

Interactive theorem provers (ITPs) are powerful tools for the formal verification of mathematical proofs down to the axiom level. However, their lack of a natural language interface remains a significant limitation. Recent advancements in large language models (LLMs) have enhanced the understanding of natural language inputs, paving the way for autoformalization—the process of translating natural language proofs into formal proofs that can be verified. Despite these advancements, existing autoformalization approaches are limited to verifying complete proofs and lack the capability for finer, sentence-level verification. To address this gap, we propose StepProof, a novel autoformalization method designed for granular, step-by-step verification. StepProof breaks down complete proofs into multiple verifiable subproofs, enabling sentence-level verification. Experimental results demonstrate that StepProof significantly improves proof success rates and efficiency compared to traditional methods. Additionally, we found that minor manual adjustments to the natural language proofs, tailoring them for step-level verification, further enhanced StepProof's performance in autoformalization.

## 1 Introduction

Mathematics is the basic tool for the development of science, and the reliability of its conclusions affects the stable growth of various disciplines. With the development of the mathematical edifice, mathematical proofs have become more complex and lengthy. The verification of mathematical proof often requires years of careful verification to ensure the accuracy of the work. However, reading a mathematical work requires a large amount of knowledge, and in the face of the many branches of mathematics today, traditional manual verification has become increasingly disastrous. Thus, an idea arose to validate mathematical work written in natural language automatically.

At present, there are two kinds of automatic verification of mathematical proof. One is to write the mathematical certificate into a machine code that can be verified by a specific expert system, which is called the interactive theorem prover (Harrison et al., 2014). After more than 40 years of development, the interactive theorem prover has begun to take shape and has been used in the verification of many mathematical works (Maric, 2015). However, because such machine-verifiable proofs need to be written in a specific programming language, whose learning cost is relatively high, interactive theorem provers are only used by a small number of mathematicians at present (Nawaz et al., 2019).

On the other hand, after the advent of the large language model (Zhao et al., 2023), through prompt engineering (Liu et al., 2023) and few shot learning (Wang et al., 2020b), large language model can be applied to many natural language processing tasks, and achieved considerable performance (Achiam et al., 2023). However, due to the hallucination problem of large language model (Ji et al., 2023), its performance in mathematics and strong logic-related work is not good enough (Huang et al., 2023), so the lack of reliability of large language model to verify mathematical work is easy to cause a lot of errors. In this context, a method that combines large language models and theorem provers gradually comes into people's view, which is called autoformalization verification (Li et al., 2024).

At present, the existing automatic formal verification work generally adopts a FULL-PROOF strategy. Although such a strategy has achieved some impressive performance in some studies, there are still many problems in the stability of its proof and the fine-grained verification. Aiming at the

problems existing in FULL-PROOF, we innovatively propose a new automatic formal proof strategy, STEP-PROOF. And the performance of StepProof is better than traditional methods in several aspects.

In summary, our contributions mainly include the following points: 1. We pioneered a novel natural language mathematical verification method StepProof, which realizes informal mathematical proof verification at the sentence level. 2. We were the first to realize the test of automatic formalization capabilities on small open-source LLMs. 3. Compared with existing methods, the StepProof method has been significantly improved in all aspects of performance.

In this paper, we will first make a brief summary of the existing relevant works and the technical background in Chapter 2. In Chapter 3, we will give a detailed introduction to the two types of strategies, FULL-PROOF and STEP-PROOF, and point out many problems faced by traditional methods. In Chapter 4, we set up a series of experiments to verify the performance of STEP-PROOF in verification tasks and its superiority over FULL-PROOF. At the same time, we also carried out a detailed analysis of some phenomena observed in the experiment to further explain the reason for the advantages. In the end, we analyze and look forward to the current limitations and future development direction.

## 2 RELATED WORK

**Theorem Prover:** Theorem provers can be roughly divided into two types, interactive theorem provers (ITPs) in which the user can enter and verify the existing proof (Asperti, 2009), and automated theorem provers (ATPs) that try to prove the statements fully automatically (Harrison, 2013). The two types of theorem provers are not mutually exclusive (Nawaz et al., 2019). Most ITPs such as Isabelle (Paulson, 1994), Coq (Huet et al., 1997) and Lean (De Moura et al., 2015) are supported by ATPs that try to automatically prove "obviously" intermediate steps in the proof entered by the user. The entered proofs are rigorously verified back to the axioms of mathematics. Different ITPs use different axiomatic foundations, e.g. set theory, first-order logic, higher-order logic, etc. Each ITP use its own language and syntax, which makes the learning cost of theorem prover high, and precludes ITPs from being widely used (Yushkovskiy, 2018).

**Large Language Model:** In recent years, large language models (LLMs) have achieved outstanding performance in many downstream tasks of natural language processing. LLMs such as Llama (Dubey et al., 2024), GPT series (Ye et al., 2023) and GLM 4 (GLM et al., 2024) are trained on large databases to understand user input in natural language and produce the related output. While large language models perform well in general tasks such as translation, their performance in dealing with logic problems has been limited. A lot of work has also shown that large language models are prone to a variety of problems when dealing with logic problems (Yan et al., 2024; Wan et al., 2024). Although in the subsequent iteration of the model, the developers provided a large amount of high-quality logic-related data to improve the logic capability of the model, the effect that could be achieved was still very limited (Lappin, 2024). Therefore, it has become a trend to add an expert system to the model to improve its accuracy, such as RAG (Fan et al., 2024), which is currently commonly used in question-answering systems. On the other hand, step-by-step reasoning has been proven can improve the reasoning ability of existing LLMs (Wei et al., 2022; Khot et al., 2022), which gives us a hint to apply in autoformalization.

**Autoformalization:** The definition of automatic formalization is very broad, but can be roughly seen as understanding and extracting translation from natural language to obtain the required structured data or formal language, such as entity relation extraction (Nguyen & Grishman, 2015). Early automatic formalization work involved extracting logical propositions from natural language in addition to entity relation extraction (Singh et al., 2020; Lu et al., 2022). However, the main problem in this kind of work is that the extracted logical propositions lack corresponding symbols for derivation and application, so the output of the automatic formalization output cannot be directly applied. With the launch of pre-trained language models such as transformer and BERT, language models have a stronger understanding ability. Wang et al. (2020a) conducted an early automatic formalization attempt for the theorem prover Mizar. However, due to the limited size and training expectations of the models at that time, the effects they could achieve were very limited. With the rapid expansion of the scale of models and predictions, many new and better performance automatic formalization work has emerged, such as the Majority Voting method by Lewkowycz et al. (2022), the DSP method by

Jiang et al. (2022)., and the method proposed by Zhou et al. (2024), in DTV (Don't Trust: Verify). They further improved the automatic formalization of the system by combining large models with some syntax-modifying filters. However, on the one hand, all these works are tested in the closed-source large model Minerva, which lacks the testing work of open source model and small model. Meanwhile, all these works adopt the strategy of FULL-PROOF, which has poor controllability for the formal output of the model, and it is difficult to locate the error point of non-formal proof. Qinghua et al. proposed a problem location method based on the FULL-PROOF strategy in SlideRule. However, this method relies heavily on the format and quality of the generated formal content, so it cannot achieve 100% problem locations detected. To solve these problems, we propose StepProof, an automatic formalization strategy that can realize sentence-level verification, to realize the verification of natural language mathematical proofs. Moreover, LEGO-Prover (Wang et al., 2023) proposed a new methodology to decompose the whole proof into several sub-proofs. Although the idea of step-proof is taking shape, it still requires some extra generation of the sub-proof formal statement generation, which increases the error probability of formalization.

## 3 STEPPROOF

In this chapter, we will provide a detailed introduction to the workflow design of StepProof. We will also compare the STEP-PROOF strategy used by StepProof with the FULL-PROOF strategy adopted by existing autoformalization systems, highlighting the problems with traditional strategies and the advantages of STEP-PROOF over FULL-PROOF.

### 3.1 FULL-PROOF

Current research on natural language proofs formal verification predominantly employs the FULL-PROOF strategy, as seen in methods like DSP and DTV. The workflow of FULL-PROOF method can be roughly illustrated as the left of Figure 1. Users submit a provable problem along with its proof process. The problem is first formalized, then the informal problem, the formalized problem, and the entire informal proof are packaged as inputs to a large language model for formalization. After obtaining the formal proof, it is combined with the formalized problem and input into a theorem prover for rule-based formal verification. The verification results are then returned to the user. Despite the clarity and simplicity of the FULL-PROOF workflow, it has significant drawbacks.

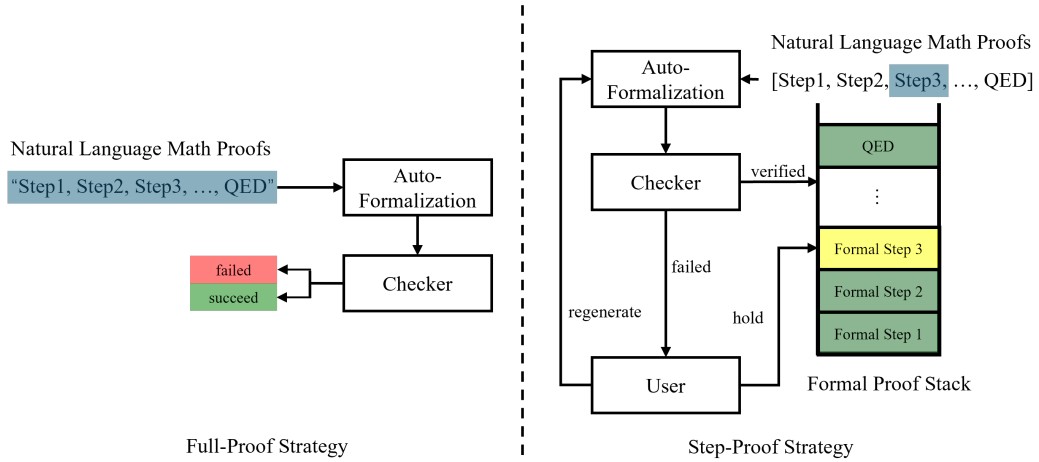

Figure 1: Full-Proof Strategy generate from the whole proof and only provide proof result instead of detailed feedback to help user improve the proof. While Step-Proof separate the whole proof into sub-proof to proof from the bottom to the top and enable the user to get more detailed feedback and fine-grained operation.

First, in the FULL-PROOF automated formalization process, the highly structured and formalized nature of the input and output, coupled with the numerous similarities in solving mathematical

equations, often leads to excessive noise in the output. To obtain the desired formalized content, numerous filters must be set up, which results in considerable waste and contamination of generated content. This can also cause generation loops, where the same content is repeatedly generated, a common issue in FULL-PROOF strategies.

Additionally, the length of proofs varies, and LLMs in FULL-PROOF struggle to adjust the output's max_new_tokens effectively based on input length. This leads to shorter proofs not being truncated in time, thus generating repetitive or noisy content, and longer proofs lacking sufficient max_new_tokens.

Lastly, the stability of FULL-PROOF generation is poor. For instance, a full proof might be almost entirely correct except for a minor error in a small step, leading to the failure of the entire formal proof. Users attempting regeneration may find previously correct parts presenting erroneous. Thus, FULL-PROOF requires highly accurate one-time generation of the entire content.

Moreover, the correlation between formal and informal content generated by FULL-PROOF is weak, making it difficult for users to map formal feedback to corresponding informal content. Although LLMs can generate formal proofs with annotations to map back to informal proofs as Qinghua et al proposed the failure step detection method in SlideRule, this approach is unstable in practice and increases the required token count.

## 3.2 STEP-PROOF

To address the numerous issues faced by the FULL-PROOF strategy, we innovatively propose STEP-PROOF. STEP-PROOF employs a step-by-step generation and verification strategy, offering better performance and stability compared to FULL-PROOF. The workflow of STEP-PROOF is illustrated in the right of Figure 1.

Unlike the one-time generation and verification of FULL-PROOF, STEP-PROOF assumes each sentence in the proof is a verifiable sub-proposition. Each step is formalized and pushed onto a formal proof stack, where it is verified along with other sub-propositions in the stack. Upon successful verification, the formalized proof and informal proof are packaged as inputs for the next step. For failed steps, the Step Proof allows users to backtrack, retaining previously verified steps and only clearing the erroneous step from the stack. The StepProof can then either re-formalize or optimize the existing informal step as needed.

STEP-PROOF offers several advantages over FULL-PROOF. First, it only needs to formalize single sentences in context, resulting in shorter, less noisy, and more stable output. The step-by-step generation strategy also means that each step's length is relatively fixed, eliminating the need for adjusting max_new_tokens and allowing the use of smaller max_new_tokens for formalizing longer theorems.

Moreover, STEP-PROOF's incremental generation and verification tolerate step errors well. Only the specific erroneous step needs to be retracted, rather than regenerating the entire content, enhancing robustness and efficiency. Finally, Step Proof ensures high correspondence between each informal and formal proof step, providing users with finer operational granularity. For instance, users can suspend a correct but incomplete step and assume it is correct to proceed, a functionality almost impossible under the FULL-PROOF strategy.

We also designed a simple and user-friendly interactive interface for the user, as shown in Figure 2. Users can complete the natural language proof of the problem through interface interaction, and each step of the proof will be formally verified after submission, and the verified proof step will be marked in green in the background to indicate that the current step has passed the verification and is reliable. If the current step does not pass the automatic validation but the user thinks it is true and wants to continue, you can select HOLD, and the current step will be highlighted in yellow in the background to indicate that the current step is in a suspended state. After completing all the proof steps, the user can input QED to indicate that the system has completed the proof, and the system will combine all the steps to perform the final verification of the proof target. Through the interactive interface, the user can also realize the PDF export of the proof process.

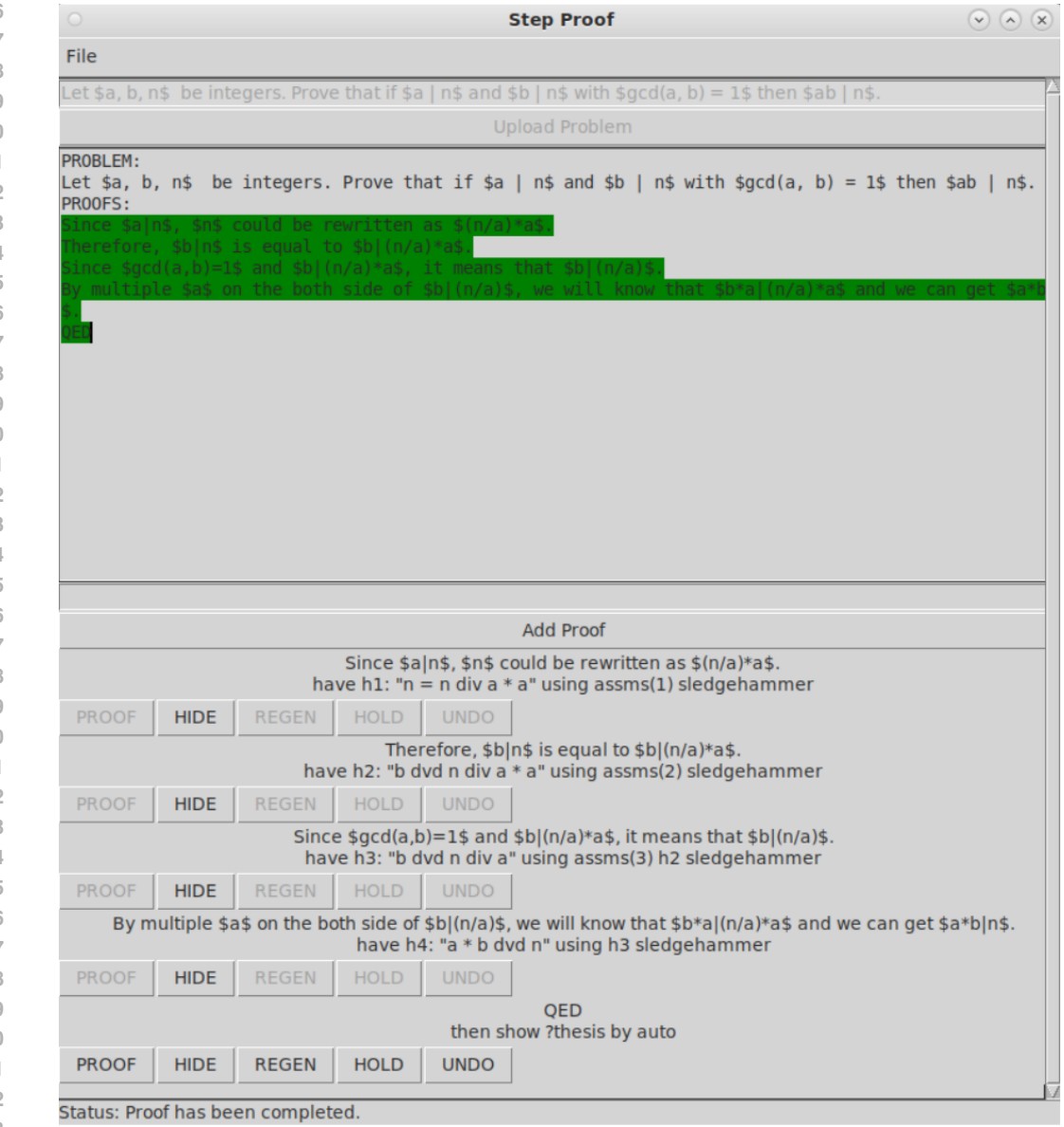

Figure 2: User Interface of StepProof

## 4 EXPERIMENT

### 4.1 EXPERIMENT SETUP

To validate the performance advantages of the STEP-PROOF strategy over FULL-PROOF, and to compare StepProof's overall performance against existing automated formal proof methods, we conducted both strategy performance tests and baseline tests using the same dataset and model settings.

For the test dataset, we used GSM8K, which includes a large number of informal mathematical problems and their correct informal proofs. These informal proofs can be easily segmented into a series of sub-steps. We chose Llama3 8B-Instruct as the large language model for automated formalization. Existing autoformalization tests use closed-source models, and we aim to fill this gap by using an open-source small LLMs.

We set the temperature to 0.3 to balance stability and flexibility. Given the small parameter count of 8B, we used a single example for the few-shot in strategy performance tests. With proof steps in GSM8K (Cobbe et al., 2021) averaging 4-5 steps, we set the max_new_tokens for FULL-PROOF to 1024, and 256 for each step in Step Proof. The test environment consisted of a single NVIDIA A4000 16GB, 8 cores of AMD5800X, and 32GB DDR4 3200 RAM. Isabelle2024 was used as the formal theorem prover with only *Main* prove library as the theorem base[1], with Isabelle-client (Shminke, 2022) as the testing service proxy.

In the strategy performance tests, we evaluated the performance of FULL-PROOF and STEP-PROOF on the GSM8K test set using the following metrics: 1. One-attempt generation proof pass rate $r_p$. 2. Average formalization time for passed proofs $\mu_f$. 3. Variance in formalization time $\sigma_f^2$. 4. Average proof time for passed proofs $\mu_p$. 5. Variance in proof time $\sigma_p^2$.

In the baseline tests, we compared the multi-attempt test results of GSM8K by Lewkowycz et al. (2022) in Majority Voting and Zhou et al. (2024) in DTV with our results. We evaluated the performance based on the number of attempts and multi-round proof pass rate, allowing up to 10 retries for each failed step in each formalization attempt.

At the same time, considering that StepProof has better granularity than traditional proof tasks, we not only evaluated the overall proof passing rate but also counted the proportion of step-proof passing. For example, if a 6-step proofs can be verified to be true in 3 steps, then we will mark that the step pass rate of the proof is 0.5. In this way, we quantify the formal proof capability of the method more comprehensively rather than the Proof Passing Rate.

In addition, to verify the influence of the writing method of non-formal proof on the passing rate of StepProof formal proof, we extracted 100 questions from the Number theory of MATH (Hendrycks et al., 2021) and made simple manual modifications to make the proof step more consistent with the proof requirement of StepProof.

## 4.2 EXPERIMENT RESULTS

|  | Proof Passing Rate | Formalization Time | Proof Time |
|---|---|---|---|
|  | $r_p$ | $\mu_f \pm \sigma_f^2$ | $\mu_p \pm \sigma_p^2$ |
| FULL-PROOF | 5.30% | 9.54 ± 12.64s | 214.93 ± 20864.97s |
| STEP-PROOF | **6.10%** | **5.83 ± 4.24s** | **130.12 ± 5271.65s** |

Table 1: Performance Test of Full-Proof and Step-Proof

In strategy performance tests, the STEP-PROOF strategy outperformed the FULL-PROOF strategy across the board on the GSM8K test set. As shown in Table 4.2, the STEP-PROOF strategy improved the one-attempt proof pass rate by **15.1%** compared to FULL-PROOF. In terms of average formalization time, STEP-PROOF required **38.9%** less time than FULL-PROOF. For average proof time, STEP-PROOF achieved a **39.5%** performance improvement over FULL-PROOF. Additionally, STEP-PROOF showed more stability in both formalization and proof time compared to FULL-PROOF. These results confirm that our strategy offers better performance, efficiency and stability.

In the baseline test as shown in Table 4.2, Step Proof surpassed DTV in multi-round verification tests on GSM8K, achieving a **10.3%** performance improvement. Moreover, StepProof required fewer attempts compared to DTV[2], demonstrating its superior proof capability and further validating the advantages of the StepProof methods.

---

[1]Here, we only use Main as the proof library, considering that the use of different libraries will greatly affect the formal verification ability of the theorem prover, in order to provide a relatively standard index. In StepProof, we do not introduce other libraries to further improve the proof ability of the theorem prover. Introducing more libraries in practice will greatly improve the proof ability of the theorem prover, and also improve the proof ability of the whole system to some extent.

[2]Don't Trust: Verify (DTV) originally used two close source models–GPT3.5 as the problem generation model and Minerva 8B as the proof generation model, while due to the Minerva being inaccessible and GPT3.5 being costing, we use the same method in DTV, but replace the LLM into Llama3.

| | Attempts | Proof Passing Rate | Comments Rate | Model |
|---|---|---|---|---|
| Majority Voting | 64 | 16.2% | - | Minerva 8B |
| Don't Trust:Verify* | 64 | 25.3% | 31.3% | Llama3 8B |
| StepProof | 10 | 22.0% | 100% | GLM4 9B(4bit) |
| StepProof | **10** | **27.9%** | **100%** | Llama3 8B |

Table 2: Baseline Test in GSM8K

| Step Pass Rate | LLAMA3 8B | | GLM4 9B(4bit) | |
|---|---|---|---|---|
| | 1 attempt | 10 attempts | 1 attempt | 10 attempts |
| $0 = r_s$ | 79.6% | 50.5% | 83.9% | 55.4% |
| $0 < r_s$ | 20.4% | 49.5% | 16.1% | 44.6% |
| $0.5 \leq r_s$ | 14.6% | 38.1% | 13.4% | 41.9% |
| $1 = r_s$ | 6.1% | 27.9% | 4.8% | 22.0% |

Table 3: Step Passing Rate Distribution in GSM8K

In the step pass rate test as shown in Table 3, we found that StepProof was able to perform some degree of validation for nearly half of the proofs after 10 rounds of trying. In LLAMA3 8B, 38.1% of the proofs completed more than half of the verification, and 27.9% of the proofs completed all of the verification. Compared with a single attempt, this is a significant improvement. At the same time, we propose a new indicator-step passing rate ($r_s$) for a more comprehensive evaluation of the proof of automatic formal verification methods.

| Step Passing Rate | Original | Modified |
|---|---|---|
| $0 = r_s$ | 35% | 32% |
| $0 < r_s$ | 65% | 68% |
| $0.5 \leq r_s$ | 42% | 45% |
| $1 = r_s$ | 6% | 12% |

Table 4: Step Passing Rate Distribution in Number Theory

In our test to verify the influence of informal proof writing on the proof pass rate (as shown in the Table 4), we found that the proof pass rate was significantly improved after simple fitting of the informal proof. This shows that compared with FULL-PROOF, STEP-PROOF adopts proof mathematics that is more suitable for step verification, which will be more conducive to improving the pass rate of automatic formal verification.

### 4.3 EXPERIMENT ANALYSIS

From Figure 3, we find that compared with the FULL-PROOF Strategy, STEP-PROOF takes less time to formalize and prove and is more stable. On the one hand, the generation strategy of Step-Proof makes the content generated in a single attempt less and more stable. Stable step content reduces the number of false proofs and the time to repeatedly prove successful content. Therefore, in terms of both generation efficiency and proof efficiency, STEP-PROOF is superior to FULL-PROOF.

To investigate the relation between step proof pass rate and number of attempts, we plotted Figure 4. It shows that most steps can be proven with relatively few attempts, with only a small fraction requiring multiple tries. We believe the main limitation to step pass rate lies not in the model's formalization ability, but in whether the informal proof steps are suitable for conversion into provable formal steps. We found that many steps in the test set cannot be formalized into provable steps. These unformalizable informal steps significantly limit the further performance improvement of

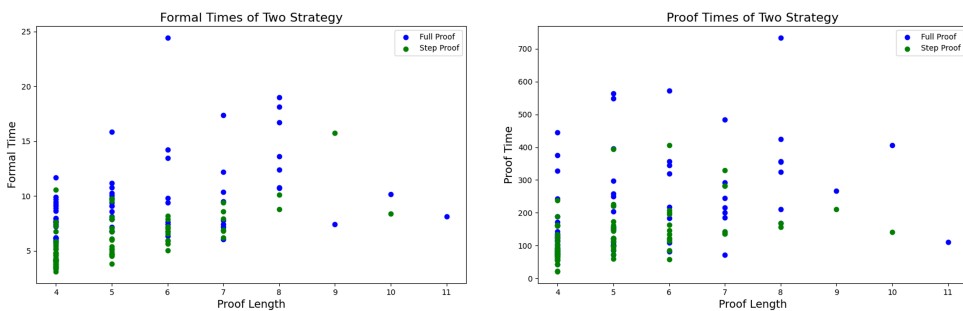

Figure 3: Time Cost of Two Strategies in Formalization and Proof

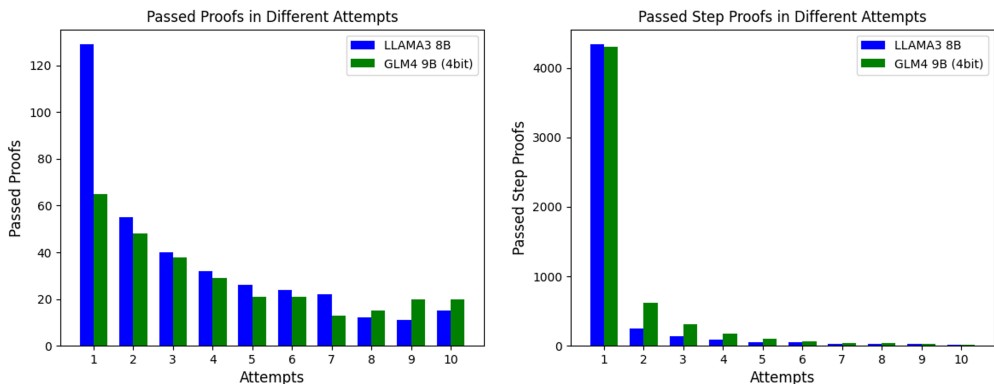

Figure 4: Passed Proofs and Steps in Different Attempts

Step Proof. Therefore, to better achieve verification in practical applications of StepProof, each proof sub-step should be ensured to be a formally verifiable sub-proposition as much as possible.

In addition, we also found that the processing capacity of LLMs is poor when faced with the processing of continuous equations, and it is easy to fall into the generation loop during the generation process, resulting in the poor quality of the generated formal proof. Therefore, in practical applications, it is necessary to split the steps as much as possible, rather than pursue once completion. At the same time, since the automatic proof ability of automated theorem proving is also limited, resolving the proof process is not only conducive to the formalization process, but also very helpful to the verification process.

## 5 LIMITATIONS & FUTURE RESEARCH

At present, the performance of StepProof in small models has been verified in various aspects. However, due to the lack of equipment and funds, the performance of StepProof in larger models has not been verified, although the previous relevant work verified the conclusion that FULL-PROOF strategy has better performance in larger models. However, the STEP-PROOF strategy has not yet been tested on a larger model.

In addition, StepProof is strict for users to enter proof steps, while FULL-PROOF is more flexible and can incorporate some non-proof explanatory language or prompt words into the proof. However, StepProof will make mistakes in the step proof because the prompt word is not provable.

Finally, StepProof pays more attention to the sequential proof with steps, but when faced with some structured proof methods, its performance is still limited, while FULL-PROOF can better capture the overall proof structure.

In the future, our work will mainly start from the following two points: 1. At present, we are writing a corpus for StepProof for automatic formal verification tasks oriented to step verification, and we hope that a more targeted corpus will help improve the step formalization ability of the model. 2. We will implement specific structured proof based on the structured design of the system to further improve the integrity and flexibility of StepProof proof.

## 6 CONCLUSION

In this paper, we innovatively propose a new automatic formal proof method called StepProof. Step-Proof implements sentence-level formal verification of natural language mathematical proofs, allowing users to conduct more flexible formal verification. At the same time, compared with the traditional FullProof strategy, StepProof has been significantly improved in formalization, proof efficiency and proof accuracy.

In addition, StepProof can preserve the contents of the proof that has already been verified, providing more information than the traditional Full-Proof strategy that can only indicate the passage and failure of the proof. We used a small model on the GSM8K data set to test the proof pass rate, and its performance reached the level of state-of-the-art.

We also conducted a test on the Number Theory dataset of MATH to test the effect of formal content writing on the proof pass rate. The experimental results show that by optimizing the non-formal proof for step verification, the passing rate of the non-formal proof can be significantly improved. We will further optimize the architecture of StepProof in future work to make it more flexible to handle formal verification of various non-formal mathematical proofs.

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

## A APPENDIX

### A.1 ADDITIONAL CASE OF MANUAL MODIFICATION FOR STEP PROOF FITTING

For the following informal problem and proof:

**Problem:** What is the average of the two smallest positive integer solutions to the congruence $14u \equiv 46 \pmod{100}$. Show it is 64.

**Solution:** Note that 14, 46, and 100 all have a common factor of 2, so we can divide it out: the solutions to

$$14u \equiv 46 \pmod{100}$$

are identical to the solutions to
$$7u \equiv 23 \pmod{50}.$$

Make sure you see why this is the case.

Now we can multiply both sides of the congruence by 7 to obtain
$$49u \equiv 161 \pmod{50},$$

which also has the same solutions as the previous congruence, since we could reverse the step above by multiplying both sides by $7^{-1}$. (We know that $7^{-1}$ exists modulo 50 because 7 and 50 are relatively prime.)

Replacing each side of $49u \equiv 161$ by a $\pmod{50}$ equivalent, we have
$$-u \equiv 11 \pmod{50},$$

and thus
$$u \equiv -11 \pmod{50}.$$

This is the set of solutions to our original congruence. The two smallest positive solutions are $-11 + 50 = 39$ and $-11 + 2 \cdot 50 = 89$. Their average is $\boxed{64}$.

In normal cases, we can cut the content according to the period to get the following sequence of non-formal proofs.

```
1  Note that $14$, $46$, and $100$ all have a common factor of $2$,
   ↪  so we can divide it out: the solutions to $$14u \equiv 46
   ↪  \pmod{100}$$ are identical to the solutions to $$7u \equiv 23
   ↪  \pmod{50}.$$
2
3  Make sure you see why this is the case.
4
5  Now we can multiply both sides of the congruence by $7$ to obtain
   ↪  $$49u \equiv 161 \pmod{50},$$ which also has the same
   ↪  solutions as the previous congruence, since we could reverse
   ↪  the step above by multiplying both sides by $7^{-1}$.
6
7  We know that $7^{-1}$ exists modulo $50$ because $7$ and $50$ are
   ↪  relatively prime.
8
9  Replacing each side of $49u\equiv 161$ by a $\pmod{50}$
   ↪  equivalent, we have $$-u \equiv 11\pmod{50},$$ and thus $$u
   ↪  \equiv -11\pmod{50}.$$
10
11 This is the set of solutions to our original congruence.
12
13 The two smallest positive solutions are $-11+50 = 39$ and
   ↪  $-11+2\cdot 50 = 89$.
14
15 Their average is $\boxed{64}$.
```

However, there are many problems with such a simple decomposition. Such as the following sentences are unable to be formalized.

```
1  Make sure you see why this is the case.
2
3  This is the set of solutions to our original congruence.
```

In addition, from the point of view of the order of proof, I can see that "*We know that $7^{-1}$ exists modulo* 50 *because* 7 *and* 50 *are relatively prime.*" is a prerequisite for "*Now we can multiply both sides of the congruence by* 7 *to obtain* $49u \equiv 161 \pmod{50}$, *which also has the same solutions as the previous congruence, since we could reverse the step above by multiplying both sides by* $7^{-1}$.", so the order of the two should be reversed.

Therefore, in order to meet the requirements of StepProof, I deleted the statements that could not be formalized and corrected the sequence to obtain the following proof sequence.

```
1  Note that $14$, $46$, and $100$ all have a common factor of $2$,
   ↪   so we can divide it out: the solutions to $$14u \equiv 46
   ↪   \pmod{100}$$ are identical to the solutions to $$7u \equiv 23
   ↪   \pmod{50}.$$
2
3  We know that $7^{-1}$ exists modulo $50$ because $7$ and $50$ are
   ↪   relatively prime.
4
5  Now we can multiply both sides of the congruence by $7$ to obtain
   ↪   $$49u \equiv 161 \pmod{50},$$ which also has the same
   ↪   solutions as the previous congruence, since we could reverse
   ↪   the step above by multiplying both sides by $7^{-1}$.
6
7  Replacing each side of $49u\equiv 161$ by a $\pmod{50}$
   ↪   equivalent, we have $$-u \equiv 11\pmod{50},$$ and thus $$u
   ↪   \equiv -11\pmod{50}.$$ This is the set of solutions to our
   ↪   original congruence.
8
9  The two smallest positive solutions are $-11+50 = 39$ and
   ↪   $-11+2\cdot 50 = 89$.
10
11  Their average is $\boxed{64}$.
```

### A.2 FORMAL CONTENT COMPARISON BETWEEN FULL-PROOF AND STEP-PROOF

**Problem:** Given that $1 < x < 2$, show that $\sqrt{(x-1)^2} + |x-2|$ can be reduced to 1.

**Solution:** Since $x > 1$, we know that $\sqrt{(x-1)^2} = x - 1$. And since $x < 2$, we know that $|x-2| = 2-x$. Therefore, it can be rewritten as $x-1+2-x$, and we know that it can be reduced to 1.

After formalize the problem we get following Isabelle statement.

```
1  theorem sqrt_abs_simplify:
2    assumes h0: "1 < x"
3    and h1: "x < 2"
4    shows "sqrt((x - 1)^2) + abs(x - 2) = 1"
```

Using Full-Proof, we can directly obtain the following formal proof.

```
1  proof -
2    have h2: "sqrt((x - 1)^2) = x - 1"
3      using h0 by (simp add: abs_of_nonneg)
4    have h3: "abs(x - 2) = 2 - x"
5      using h1 by (simp add: abs_of_nonpos)
6    have "sqrt((x - 1)^2) + abs(x - 2) = (x - 1) + (2 - x)"
7      using h2 h3 by simp
8    also have "... = 1"
9      by simp
10   finally show ?thesis .
11  qed
```

In the StepProof approach, we would firstly input the first step

```
1  Since $x > 1$, we know that $\sqrt{(x-1)^2} = x - 1$.
```

Then after formalization, we can get corresponding formal proof.

```
1  have h2: "sqrt((x - 1)^2) = x - 1" using h0 by simp
```

After this formal step has been verified, we will input the next steps in the same way.

```
1  And since $x < 2$, we know that $|x-2| = 2-x$.
```

```
1  have h3: "abs(x - 2) = 2 - x" using h1 by simp
```

```
1  Therefore, it can be rewritten as $x - 1 + 2 - x$, and we know
   ↪   that it can be reduced to 1.
```

```
1  show ?thesis using h2 h3 by simp
```

After the global goal has been proven, the system will automatically align the formal proof with the informal proof. Then we will get the over all formal proof of StepProof as following

```
1  proof-
2          (*Since $x > 1$, we know that $\sqrt{(x-1)^2} = x - 1$.*)
3          have h2: "sqrt((x - 1)^2) = x - 1" using h0 by simp
4          (*And since $x < 2$, we know that $|x-2| = 2-x$.*)
5          have h3: "abs(x - 2) = 2 - x" using h1 by simp
6          (*Therefore, it can be rewritten as $x - 1 + 2 - x$, and
   ↪   we know that it can be reduced to 1.*)
7          show ?thesis using h2 h3 by simp
8
9  qed
```

