# Reviewer nEet

# Weaknesses

## Weakness 1

**Comment:** Unsubstantiated Claim of Advantage: The claimed advantage of StepProof's selective error correction (where only erroneous steps are retracted rather than the entire proof) is not unique to StepProof. Interactive theorem provers (ITPs) inherently support stepwise correction, enabling users to fix specific errors without requiring a full retraction. Thus, StepProof's advantage in this aspect appears overstated.

**Response:** The main contribution of StepProof lies in step-by-step formal generation and distributed verification. Step-by-step verification is indeed a characteristic of Isabelle as an ITPs, as you say. However, StepProof focuses on achieving step-by-step formal verification at the natural language level. This implementation is significantly different from existing natural language formalization and verification methods. Users can clearly locate the steps that prove failed, and make more flexible adjustments and proof operations such as (hold). This is not possible with existing natural language formal verification methods.

## Weakness 2

**Comment:** Lack of Novelty in Stepwise Translation: Stepwise translation in autoformalization is not a new concept. Previous methods, like the DSP approach, have already implemented similar methodologies. These methods translate decomposed proof steps, whether generated by an LLM or provided by a human, indicating that StepProof may not be as innovative as claimed in this area.

**Response:** There is no doubt that the DSP does implement automatic formal mathematical verification. Our team has studied and tried to reproduce DSP code, so we know the specific implementation of DSP. The formalization way of DSP still adopts the Full Proof strategy described in our article. The emphasis of DSP is to replace the verification methods of the formal steps by using different heuristics for each step of the generated formal proof. This is completely

different from what we are trying to do at the natural language level to formalize and prove the steps. We have tried to locate failed natural language steps and enhance the stability of generated proofs under DSP framework, which is very difficult, but it is very easy under StepProof framework. Therefore, we have reasons to believe that StepProof and DSP have essential differences in function and implementation.

## Weakness 3

**Comment:** Overly Restrictive Assumptions: StepProof's framework assumes that each sentence in a proof can be treated as an independent, verifiable sub-proposition, which limits its applicability. This subgoal-based approach does not align well with many natural logical structures in proofs, especially those involving complex logical dependencies or sequence reordering. As a result, StepProof might require significant manual adjustments for compatibility with common proof structures.

**Response:** We agree with your opinion on the limitations of StepProof, which has also been mentioned in our limitation in the article. Focusing on the details of the verification will inevitably lead to a certain lack of overall content. This will be the focus of our future work. However, it is worth noting that the current global generation will inevitably face the problems of ignoring details and lacking logic in the case of long proof. At present, there is no perfect solution that can balance the integrity and attention to detail. At present, there is no work focusing on such details as step verification. Therefore, we believe that although StepProof does have such problems, as an attempt to the details, there is nothing wrong with such problems.

## Weakness 4

**Comment:** Ambiguous Evaluation Methodology: The incremental verification of sub-propositions might count intermediary, potentially incorrect and mathematically misleading results as "correct" sub-proofs, leading to an inaccurate reflection of the method's overall success. This evaluation ambiguity calls into question the validity of the reported improvements in success rates.

**Response:** Given isabelle's consistency in context, step validation does not lead to the situation you mentioned. As for the success rate mentioned. It is possible in some cases, but this is based on the premise of artificially adding irrelevant steps. StepProof is intended to be applied as an interactive program, rather than an automated detection system, and used to score higher. We use the step passing rate to see how far it can go, not how well it can do.

# Weakness 5

**Comment:** Inappropriate Benchmark Dataset: The authors' choice of GSM8K as a benchmark dataset is unsuitable for evaluating proof autoformalization due to its relative simplicity and lack of complex logical structures. Datasets like ProofNet or MiniF2F would provide a more accurate measure of StepProof's performance on challenging, real-world mathematical proofs, better reflecting its practical value in formalization tasks.

**Response:** We are also very much looking forward to using ProofNet and MiniF2F as test datasets. However, it is important to note that our work focuses on the verification of "human-written proofs." We don't expect LLMS to be strong enough to understand complex logic. In fact, we don't think LLM can solve complex logic problems. We use LLM more as a translation software to translate informal content into formal content. MiniF2F does not contain human-written proofs, so it is not a suitable dataset for our testing. As for ProofNet, it does meet our needs for human written proofs, but we wish you could consider that we are not rushing to deal with these strong logic problems because we don't think LLM is strong enough for logical reasoning. What's more, limited to our hardware devices (single A4000), we can only use small models such as 8B to test, and its performance may be terrible. Most importantly, we need to know that automatic formalization is not 100% correct even in a simple GSM8K. Why should we rush to try more difficult problems. It's like, I can do the Olympiad math, but I can't do all the primary school math correctly. Do you think that makes sense?

# Weakness 6

**Comment:** Insufficient Detail on Methodology: It is also worth noting that further clarification on the prompting and output syntax for both StepProof and

Full-Proof in the comparison experiments is needed. More specific details on the handling of the LLM's guessed proof states could be valuable for assessing the validity and replicability of the reported performance differences between the two approaches.

**Response:** We are sorry about that. Limited by space, we didn't provide enough detail as you wanted in the article. On this point, we will add more details about this aspect in the appendix in the later revision work.

# Questions

## Question 1

**Comment:** Consider Additional Benchmark Datasets: Suggest that the authors include additional, more complex datasets like ProofNet or MiniF2F in future evaluations.

**Response:** We will actively consider including these two datasets for testing in the future work. However, due to the current conditions and the long experiment period, we will consider making supplements in the future related work.

## Question 2

**Comment:** Highlight Distinctions from Prior Work on Stepwise Translation: Advise the authors to address the similarities between StepProof and prior stepwise translation methods.

**Response:** We think the response has been metioned in the reponse of Weakness 2.

## Question 3

**Comment:** How does StepProof handle complex proof structures outside the subgoal-based framework? Could the authors elaborate on StepProof's limitations in handling complex, non-linear proof structures? For example, how

does it manage proofs that involve nested assumptions, indirect reasoning, or statements that need reordering for coherence?

**Response:** Please refer to the response on Weakness 3.

# Question 4

**Comment:** How does StepProof compare to Whole-Proof when using more informative LLM feedback? In Whole-Proof, does the LLM output proof states after each line, or is this feature limited to StepProof? Allowing Whole-Proof access to these proof states could potentially improve its success rate. Could the authors provide more details on the prompting strategies for both methods?

**Response:** In our context, we metioned that existing Whole-Proof method can only show if the Whole-Proof can be proved succeed or not. StepProof firstly realize the sentence level proof states check. We will provide more details on the prompting strategies for both method in the appendix.

# Reviewer EFnZ

# Weaknesses

# Weakness 1

**Comment:** It is convincing that the step-by-step reasoning works well, but I feel it natural and unsurprising because it has been proven that LLMs can better carry out step-by-step logical reasoning than one-step [1]. Thus, I'm unsure how significant the contribution of the paper, which seems to confirm step-by-step proof generation works more well than one-step generation, is.

**Response:** We agree with you that the logical reasoning of the steps can lead to better results has been raised. But this has not been tested and validated in automatic formal validation. One of our contributions is that we were the first to try this in automatic formal validation. Secondly, the automatic formalization of step verification not only provides more excellent formalization ability from the LLM side, but the most important thing is that step generation combines the

new characteristics and more refined operation mode realized by the theorem prover. These new features are not available in the existing overall generation strategy.

# Weakness 2

**Comment:** Furthermore, it seems that the paper is not the first to apply the step-by-step reasoning power of LLMs to formal proof generation. For example, LEGO-Prover [2] decomposes informal proofs into step-by-step informal proofs with sub-goals and then proves the generated sub-goals. Although the main aim of LEGO-Prover is to address growing libraries, the paper does not theoretically, empirically, qualitatively, nor quantitatively compare the proposed approach with such existing approaches that exploit the step-by-step reasoning ability of LLMs.

**Response:** LEGO-PROVER focuses more on tactics traceability. StepProof focuses more on the detection of the correctness of the step, which is not a small difference. In addition, LEGO-PROVER needs to decompose formal-goals, which requires other new formal-statements to be generated. And because LLM is not good at generating formal-statements, it is also easy to introduce new errors. In StepProof, a single formal-statement ensures the consistency of the proof.

# Questions

# Question 1

**Comment:** Is it possible to explain, prove, and/or demonstrate how the application of step-by-step reasoning in proof generation differs from or advances beyond the previous work like [1,2]?

**Response:** The responses had been clarified in Weakness 1 and Weakness 2.

# Question 2

**Comment:** Table 2: What's "Comments Rate"?

**Response:** Comments Rate is an indicator about the generated formal proof include corresponsed comments with informal proofs. SlideRule propose a method to indicate the failed informal step. In StepProof, our methodology could let the formal proof align to the informal proof with 100%.

# Question 3

**Comment:** Table 3: $r_s$ is a step pass rate?

**Response:** Yes $r_s$ is the step pass rate.

# Question 4

**Comment:** "simple fitting" Is the fitting necessary? It seems to mean one needs to tune the proofs.

**Response:** Fitting is not necessary, this part of the study is to show that fitting can provide better performance and results.

# Reviewer rYkN

# Weaknesses

## Weakness 1

**Comment:** Decomposition approach limitations. I am not sure about your approach of decomposing the informal proof into independent subpropositions. "STEP-PROOF assumes each sentence in the proof is a verifiable sub-proposition" - are you really just breaking by syntactic checks for sentences? What if you have a subproposition that is expressed in multiple sentences with dependencies or contextual information between them? Perhaps a better approach would be to try to use the LLM to explicitly and more intelligently decompose the informal proof into independent sub-propositions (or lemmas) as is commonly done in compositional approaches with LLMs. See e.g. (Tushar et al ICLR'23, Pourreza et al NeurIPS'23). This is particularly concerning: "and made

simple manual modifications to make the proof step more consistent with the proof requirement of StepProof". Firstly, this shows that StepProof is not directly capable of handling arbitrary informal NL proofs. Secondly, you can provide much more details here in terms of what manual modifications were required (you have plenty space left in the page limit and unlimited space in the appendix). You can explain general classes of modifications that needed to be made, and also provide many samples of the modifications you made to help the reader judge how "simple" the modifications are.

**Response:** In StepProof, the system considers each input by the user as a verifiable proof step. Rather than receiving all the proofs at once and decompose the proofs. Therefore, in StepProof, the user's submission does not have to be one sentence at a time. The role of StepProof is to help users with step verification, so how to decompose the non-formal proof is up to the user. For example, if the user's first proof is wrong, there is no point in subsequent proofs being correct. Instead of finding all the wrong steps in the user's entire proof, StepProof helps the user build a reliable, validated formal proof. This process is like a stack process. We are sorry that the manual modification part does not provide enough details, and we will add these relevant information in the appendix later.

# Weakness 2

**Comment:** Evaluation limitations. Though showing core value to some degree, the main evaluation results do not show a very strong improvement (6.1% vs 5.3% on compositional vs direct strategy and 27.9% vs 25.3% in comparison with the best DTV baseline). These seem pretty marginal and may be within margin of random variations in experiments and LLM performance. Also, only one dataset (GSM8K) is used - not sure if this shows generality of the approach, especially given its assumptions of decomposition at the sentence level which would be good to test on more datasets. The number of attempts comparison between StepProof and baselines is interesting - 10 vs 64 attempts is a significant improvement for step proof. But can you clarify: are these the settings of the attempts parameter that you have chosen? Did the baselines actually require this many attempts or was their performance similar with fewer attempts? Perhaps a more explicit investigation of this would be helpful - e.g. a graph showing how the performance (accuracy) of both your system and baselines (y axis) increases or changes with the number of attempts (x axis).

**Response:** 64 is the number of attempts used in the two baseline works (it can be found in their papers). The performance increased from several aspects not only limited to the accuracy but also include the efficiency. We are sorry that the test dataset is only GSM8K. However, StepProof is a little different from previous formalization work. StepProof's work is not mainly focused on improving the performance of automatic formalization (although it has improvement over previous work). The most important point of StepProof is that the step verification enables the user to locate the informal proof step that fails to verify. Then user can make more refined operations and adjustments. We also explained in our response of Weakness 1 that StepProof is not used to decompose the user's complete proof. Instead, the user submit each step based on their requirement, then system will formalize and verify the step, then user submits the next step and so on until finally builds a whole proof whose all the steps have been verified. Since there is no such dataset that writing proof in a step way. Therefore, we have to decompose the proof into steps and test them. The main reason we didn't choose MATH as our test dataset was because we didn't have an automated way to cut a complete proof into steps, so we didn't use MATH as our test dataset. On the other hand, there is no non-formal proof in MiniF2F, so we did not choose MiniF2F. The main reason for choosing GSM8K is because we observed that in most cases in GSM8K, each sentence can be regarded as a subproof step, so we finally chose GSM8K as our test dataset. What's more, after manual segmentation with some proofs in the Number Theory of MATH, we get some result about MATH, then after the deletion and correction of some non-proof steps, the table of manual modification was finally obtained. Since our equipment is limited, with only one A4000, the models we can test are limited. And the period of automatic formal verification test is very long, so we have to consider more points in the selection of test dataset.

# Weakness 3

**Comment:** Presentation problems. Many errors, inconsistencies and presentation/organization issues make the paper difficult to read and follow. Please improve upon these. Some examples:

- "The workflow of STEP-PROOF is illustrated in the left of Figure 1." - should be in the right
- E.g., "As shown in Table 4.2" should be "Table 1" and then again for the baselines is states "In the baseline test as shown in Table 4.2"! which should

be "Table 2". Please check for such mistakes and organize the paper better.

- many references do not state the conferences where the papers have already been accepted - please improve the quality of references
- organization of the paper in terms of sectioning needs much improvement - especially in the evaluation section, you seem to switch focus to different aspects of evaluation abruptly in the next paragraph and it is pretty confusing - can you please organize different aspects into appropriate subsections (you seem to have plenty of space left in the page limit anyway).
- In the first paragraph of evaluation you state "we conducted both strategy performance tests and baseline tests" - please clarify that what you mean by strategy performance tests and what you mean by baseline tests - it took a while to understand what these meant after back and forth reading and it helps to clearly introduce the goals of the evaluation to the user in normal language without paper-specific terminology.
- please clarify in Table 1 that the proof pass rate is for ONE ATTEMPT and that in Table 2 it is for multiple attempts (it took a while to understand why the proof pass rate was low here as compared to TABLE 2).

**Response:** In fact, there are two sets of experiments here. One is to compare the performance differences between Full-Proof and Step-Proof in various aspects, and the performance difference is tested under one attempt. The multi-attempt test is compared with the baseline, comparing the performance under multiple attempts, of course, multi-attempt testing must also contain one attempt, but because there is no record of time spent in multiple rounds of testing, so the two will have some similarities but not same. Since there is some randomness in One Attempt, it may appear lower than in TABLE2. In fact, if you look at the left side of Figure 4, it is clear that the pass rate in one attempt is not low, but since the multi-attempt does not include the record of time, so we didn't use that pass rate.

# Questions

## Question 1

**Comment:** Are all evaluations you have done with StepProof in fully automated manner? So there are no user interactions in these evaluations? Please clarify

that if its the case.

**Response:** We have mentioned this part in my response on Weakness 2. Hope it can solve your question. In short, because the proofs in GSM8K are suitable for automated decomposition, we automatically decompose the proofs in GSM8K for automated testing. Since proofs in MATH are difficult to decompose automatically, we manually decompose some of the proofs and perform manual modification tests.

# Question 2

**Comment:** What is "comments rate" in table 2? Is it the amount of feedback from the verification system? What does the 100% for StepProof and 31.3% for DTV mean? Please include some discussion of this and how exactly it may be relevant.

**Response:** The Comments Rate is a metric mentioned in a piece of work called SlideRule. This metric is used to estimate the proportion of comments that contain non-formal proofs in the formal proofs generated. SlideRule realizes the localization detect of the informal content of the problem by including informal proof comments in the formal proof. However, since not every formal proof generated contains non-formal comments, this method of problem positioning is unstable, and StepProof solves this problem mechanically, so its Comment Rate is 100%.

# Question 3

**Comment:** Why did you limit the number of attmpts in step proof to be 10? (while other methods like DTV you have up to 64 attempts). What happens after 10 attempts? Does the system improve further, or gains diminish, or could there even be any degradation as well?

**Response:** We think we have given some answers to this point in our reply to Weakness 2. On the one hand, due to the limitations of the equipment, the test period of our trials is extremely long, so we cannot afford more rounds of testing. Secondly, we think Figure 4 also explains this question. From Figure 4, we can see that most proofs and steps can succeed in fewer attempts, while the proportion of proofs requiring more rounds of attempts to succeed is very small.

Therefore, even if we increase the number of attempts to 64, the improvement will be negligible. So we think 10 times is enough.

# Question 4

**Comment:** Can StepProof cause worse performance if the proof fails due to sentence level decomposition problems, while the fullproof may still work as it has the whole context?

**Response:** In some cases, the wrong use of StepProof will lead to its performance is not as good as Full-Proof. But the comparison is unfair. Just like driving in traffic jams, it will be slower than riding a bicycle, but this does not mean that the car is slower than the bicycle. More importantly, the core of StepProof is that it allows the user to locate the problematic step and gives the user the possibility to modify it. At the same time, StepProof allows to keep the correct part and modify the wrong part, which is impossible for the Full-Proof. Full-Proof can not keep the correct content, only the wrong content is modified while StepProof has better stability in this respect.