# OpenReview forum: "StepProof: Step-by-step verification of natural language mathematical proofs"
_ICLR.cc/2025/Conference — Submitted to ICLR 2025_

### Official Review · Reviewer_rYkN · 2024-10-23

**Soundness:** 2
**Presentation:** 2
**Contribution:** 2
**Rating:** 6
**Confidence:** 3

**Summary:**

This paper proposes a method for step-by-step verification of natural language mathematical proofs. Unlike traditional "full-proof" approaches, which formalize and verify entire proofs at once, the proposed method ("StepProof") breaks down proofs into smaller, sentence-level subproofs to perform more granular verification. This compositional approach also allows for an intereactive user experience as well as error handling by allowing verification of individual steps and retrying only failed steps without discarding the entire proof. An evaluation is shown that the StepProof strategy performs better than a full-proof strategy and two existing baseline approaches for auto formalization on the GSM8K dataset.

**Strengths:**

1. The compositional verification approach is sensible and seems relatively novel in the autoformalization domain (though similar approaches have been explored in many other domains). Its benefits are intuitive and clear as compared to generating entire proofs at once (detecting errors, not redoing entire proofs, increased robustness)
2. The experimental results do show core utility of the ideas with improvements over full proof strategy and baselines in terms of proof success/number of attempts, though there is much room for improvement of evaluation in various aspects. Bringing the study to use open source models like Llama and implementing baseline systems here is also admirable and can help progress the research area with broader accessibility.
3. The compositional approach not only has improved results for full automation, but also creates a foundation for a more interactive user experience (more fine grained feedback from steps verification, allowing user to change or improve individual steps or skip steps in the proof, etc). Though this aspect has not been directly evaluated with real users in this work, it is a good direction to take in the autoformalization domain to provide more control and assistance for users.

**Weaknesses:**

1. Decomposition approach limitations. I am not sure about your approach of decomposing the informal proof into independent subpropositions. "STEP-PROOF assumes each sentence in the proof is a verifiable sub-proposition" - are you really just breaking by syntactic checks for sentences? What if you have a subproposition that is expressed in multiple sentences with dependencies or contextual information between them? Perhaps a better approach would be to try to use the LLM to explicitly and more intelligently decompose the informal proof into independent sub-propositions (or lemmas) as is commonly done in compositional approaches with LLMs. See e.g. (Tushar et al ICLR'23, Pourreza et al NeurIPS'23). This is particularly  concerning: "and made simple manual modifications to make the proof step more consistent with the proof requirement of StepProof". Firstly, this shows that StepProof is not directly capable of handling arbitrary informal NL proofs. Secondly, you can provide much more details here in terms of what manual modifications were required (you have plenty space left in the page limit and unlimited space in the appendix). You can explain general classes of modifications that needed to be made, and also provide many samples of the modifications you made to help the reader judge how "simple" the modifications are.
2. Evaluation limitations. Though showing core value to some degree, the main evaluation results do not show a very strong improvement (6.1% vs 5.3% on compositional vs direct strategy and 27.9% vs 25.3% in comparison with the best DTV baseline). These seem pretty marginal and may be within margin of random variations in experiments and LLM performance. Also, only one dataset (GSM8K) is used - not sure if this shows generality of the approach, especially given its assumptions of decomposition at the sentence level which would be good to test on more datasets. The number of attempts comparison between StepProof and baselines is interesting - 10 vs 64 attempts is  a significant improvement for step proof. But can you clarify: are these the settings of the attempts parameter that you have chosen? Did the baselines actually require this many attempts or was their performance similar with fewer attempts? Perhaps a more explicit investigation of this would be helpful - e.g. a graph showing how the performance (accuracy) of both your system and baselines (y axis) increases or changes with the number of attempts (x axis).
3. Presentation problems. Many errors, inconsistencies and presentation/organization issues make the paper difficult to read and follow. Please improve upon these. Some examples:
- "The workflow of STEP-PROOF is illustrated in the left of Figure 1."  - should be in the right
- E.g., "As shown in Table 4.2" should be "Table 1" and then again for the baselines is states "In the baseline test as shown in Table 4.2"! which should be "Table 2". Please check for such mistakes and organize the paper better.
- many references do not state the conferences where the papers have already been accepted - please improve the quality of references
- organization of the paper in terms of sectioning needs much improvement - especially in the evaluation section, you seem to switch focus to different aspects of evaluation abruptly in the next paragraph and it is pretty confusing - can you please organize different aspects into appropriate subsections (you seem to have plenty of space left in the page limit anyway).
- In the first paragraph of evaluation you state "we conducted both strategy performance tests and baseline tests" -  please clarify that what you mean by strategy performance tests and what you mean by baseline tests - it took a while to understand what these meant after back and forth reading and it helps to clearly introduce the goals of the evaluation to the user in normal language without paper-specific terminology.
- please clarify in Table 1 that the proof pass rate is for ONE ATTEMPT and that in Table 2 it is for multiple attempts (it took a while to understand why the proof pass rate was low here as compared to TABLE 2)


Referenced Related works:
- Decomposed Prompting: A MODULAR APPROACH FOR SOLVING COMPLEX TASKS
Tushar Khot, Harsh Trivedi, Matthew Finlayson, Yao Fu,Kyle Richardson, Peter Clark, Ashish Sabharwal. In ICLR 2023
- Mohammadreza Pourreza and Davood Rafiei. Din-sql: decomposed in-context learning of text-to sql with self-correction. In NeurIPS 2023

**Questions:**

1. Are all evaluations you have done with StepProof in fully automated manner? So there are no user interactions in these evaluations? Please clarify that if its the case.
2. Also, some evaluation of interactive features would be good, e.g. with user studies.
3. What is "comments rate" in table 2? Is it the amount of feedback from the verification system? What does the 100% for StepProof and 31.3% for DTV mean? Please include some discussion of this and how exactly it may be relevant.
4. Why did you limit the number of attmpts in step proof to be 10? (while other methods like DTV you have up to 64 attempts). What happens after 10 attempts? Does the system improve further, or gains diminish, or could there even be any degradation as well?
5. Can StepProof cause worse performance if the proof fails due to sentence level decomposition problems, while the fullproof  may still work as it has the whole context?

---

### Official Review · Reviewer_pXFu · 2024-11-03

**Soundness:** 1
**Presentation:** 1
**Contribution:** 1
**Rating:** 1
**Confidence:** 5

**Summary:**

This paper describes a framework for granular, step-by-step verification of natural language reasoning.

**Strengths:**

N.A.

**Weaknesses:**

- The paper is almost impossible to comprehend. I highly suspect a big portion of it is generated by LLMs.
- The full-proof strategy so baffling: I am not even sure what is formal backend and what is role of users here.

**Questions:**

N.A.

---

### Official Review · Reviewer_EFnZ · 2024-11-04

**Soundness:** 2
**Presentation:** 1
**Contribution:** 1
**Rating:** 3
**Confidence:** 4

**Summary:**

The paper proposes generating formal proofs from informal ones in a step-by-step manner and shows that it outperforms one-time proof generation.

**Strengths:**

+ The paper applies the known idea that, for LLMs, the step-by-step logical reasoning works more well than one-step reasoning, to autoformalization, especially the formal proof generation.

**Weaknesses:**

- It is convincing that the step-by-step reasoning works well, but I feel it natural and unsurprising because it has been proven that LLMs can better carry out step-by-step logical reasoning than one-step [1]. Thus, I'm unsure how significant the contribution of the paper, which seems to confirm step-by-step proof generation works more well than one-step generation, is.
- Furthermore, it seems that the paper is not the first to apply the step-by-step reasoning power of LLMs to formal proof generation. For example, LEGO-Prover [2] decomposes informal proofs into step-by-step informal proofs with sub-goals and then proves the generated sub-goals. Although the main aim of LEGO-Prover is to address growing libraries, the paper does not theoretically, empirically, qualitatively, nor quantitatively compare the proposed approach with such existing approaches that exploit the step-by-step reasoning ability of LLMs.

[1] Jason Wei, Xuezhi Wang, Dale Schuurmans, Maarten Bosma, Brian Ichter, Fei Xia, Ed H. Chi, Quoc V. Le, Denny Zhou: Chain-of-Thought Prompting Elicits Reasoning in Large Language Models. NeurIPS 2022

[2] Haiming Wang, Huajian Xin, Chuanyang Zheng, Zhengying Liu, Qingxing Cao, Yinya Huang, Jing Xiong, Han Shi, Enze Xie, Jian Yin, Zhenguo Li, Xiaodan Liang:
LEGO-Prover: Neural Theorem Proving with Growing Libraries. ICLR 2024

Other comments and concerns

- Please insert spacing before citations, like "the large language model(Zhao et al., 2023)" --> "... model (Zhao et al., 2023)" in line 44. I suggest reviewing all the citations to avoid similar issues.
- l88 "A lot of work has also shown that ..." Please cite the papers showing it.
- l100 "bert" --> BERT?
- l106 "DTV" Please explain the abbreviation where it appears first.
- l110 "Qinghua et al." No link to the citation.
- l155-160: It would be nice to explain the problem with concrete examples that enable the reader to easily find the issue in FULL-PROOF strategies.
- l161 "max_new_tokens" No explanation about this.
- l338 "new index" I can't find what this is.

**Questions:**

- Is it possible to explain, prove, and/or demonstrate how the application of step-by-step reasoning in proof generation differs from or advances beyond the previous work like [1,2]?
- Table 2: What's "Comments Rate"?
- Table 3: $r_s$ is a step pass rate?
- l351 "simple fitting" Is the fitting necessary? It seems to mean one needs to tune the proofs.

---

### Official Review · Reviewer_nEet · 2024-11-04

**Soundness:** 3
**Presentation:** 3
**Contribution:** 2
**Rating:** 3
**Confidence:** 4

**Summary:**

The paper introduces StepProof, a method designed to improve the verification of natural language mathematical proofs by breaking them down into smaller, verifiable subproofs. Unlike traditional autoformalization methods that only verify complete proofs, StepProof operates on a sentence level, enabling granular verification of each step. This approach aims to enhance the efficiency and accuracy of proof verification by targeting and resolving errors within individual steps, rather than regenerating entire proofs. Experimental results indicate that StepProof outperforms other approaches in proof success rates and efficiency. Minor manual modifications to the proofs, aligning them with step-level requirements, were also found to enhance the performance of StepProof.

**Strengths:**

- Granular Verification Approach: StepProof’s method of breaking down proofs into verifiable sub-propositions is a notable advancement compared to traditional methods that require verifying complete proofs. This could allow for a more targeted approach to error correction by addressing specific subproofs.
- Improved Success Rates and Efficiency: The experimental findings indicate that StepProof improves both success rates and efficiency for proof verification over baseline methods. By enabling sentence-level verification, StepProof appears to make the autoformalization process more manageable and less resource-intensive, which could be beneficial for large-scale applications.
- Enhanced Usability: The authors found that the minor adjustments to natural language proofs tailored for step-level verification, which was made easier by the GUI interfaces, further enhance StepProof’s performance, suggesting practical guidance for users to maximize the method's effectiveness.

**Weaknesses:**

- Unsubstantiated Claim of Advantage: The claimed advantage of StepProof’s selective error correction (where only erroneous steps are retracted rather than the entire proof) is not unique to StepProof. Interactive theorem provers (ITPs) inherently support stepwise correction, enabling users to fix specific errors without requiring a full retraction. Thus, StepProof’s advantage in this aspect appears overstated.
- Lack of Novelty in Stepwise Translation: Stepwise translation in autoformalization is not a new concept. Previous methods, like the DSP approach, have already implemented similar methodologies. These methods translate decomposed proof steps, whether generated by an LLM or provided by a human, indicating that StepProof may not be as innovative as claimed in this area.
- Overly Restrictive Assumptions: StepProof’s framework assumes that each sentence in a proof can be treated as an independent, verifiable sub-proposition, which limits its applicability. This subgoal-based approach does not align well with many natural logical structures in proofs, especially those involving complex logical dependencies or sequence reordering. As a result, StepProof might require significant manual adjustments for compatibility with common proof structures.
- Ambiguous Evaluation Methodology: The incremental verification of sub-propositions might count intermediary, potentially incorrect and mathematically misleading results as “correct” sub-proofs, leading to an inaccurate reflection of the method's overall success. This evaluation ambiguity calls into question the validity of the reported improvements in success rates.
- Inappropriate Benchmark Dataset: The authors’ choice of GSM8K as a benchmark dataset is unsuitable for evaluating proof autoformalization due to its relative simplicity and lack of complex logical structures. Datasets like ProofNet or MiniF2F would provide a more accurate measure of StepProof’s performance on challenging, real-world mathematical proofs, better reflecting its practical value in formalization tasks.
- Insufficient Detail on Methodology: It is also worth noting that further clarification on the prompting and output syntax for both StepProof and Full-Proof in the comparison experiments is needed. More specific details on the handling of the LLM’s guessed proof states could be valuable for assessing the validity and replicability of the reported performance differences between the two approaches.

**Questions:**

- Consider Additional Benchmark Datasets: Suggest that the authors include additional, more complex datasets like ProofNet or MiniF2F in future evaluations.
- Highlight Distinctions from Prior Work on Stepwise Translation: Advise the authors to address the similarities between StepProof and prior stepwise translation methods.
- How does StepProof handle complex proof structures outside the subgoal-based framework?
Could the authors elaborate on StepProof’s limitations in handling complex, non-linear proof structures? For example, how does it manage proofs that involve nested assumptions, indirect reasoning, or statements that need reordering for coherence?
- How does StepProof compare to Whole-Proof when using more informative LLM feedback?
In Whole-Proof, does the LLM output proof states after each line, or is this feature limited to StepProof? Allowing Whole-Proof access to these proof states could potentially improve its success rate. Could the authors provide more details on the prompting strategies for both methods?

---

### Meta-Review · Area_Chair_7Jdd · 2024-12-18

**Metareview:**

The reviewers stated that defining verifiable sub-propositions of proofs may pose an advantage compared to traditional proof verification methods. The experimental results show potential superiority over baseline methods.

However, the advantages over other methods are not clear, and the lack of novelty could not be disputed by the authors. Among other points, the reviewers state very restrictive assumptions and that more details on the evaluation are needed.

**Additional Comments On Reviewer Discussion:**

Since the authors did not reply to the reviews, the discussion led to a clear rejection of this paper.

---

### Decision · Program_Chairs · 2025-01-22

Reject